# Value of laboratory results in addition to vital signs in a machine learning algorithm to predict in-hospital cardiac arrest: A single-center retrospective cohort study

Ryo Ueno [1,2,3]*, Liyuan Xu[4], Wataru Uegami[5], Hiroki Matsui[6], Jun Okui [7], Hiroshi Hayashi[7], Toru Miyajima[7], Yoshiro Hayashi[1], David Pilcher[2], Daryl Jones[2,3]

1 Department of Intensive Care Medicine, Kameda Medical Center, Chiba, Japan, 2 Australian and New Zealand Intensive Care Research Center, School of Public Health and Preventive Medicine, Monash University, Melbourne, Australia, 3 Department of Intensive Care, Austin Hospital, Melbourne, Australia, 4 Department of Computer Science, Graduate School of Information Science and Technology, The University of Tokyo, Tokyo, Japan, 5 Anatomical Pathology, Kameda Medical Center, Chiba, Japan, 6 Clinical Research Support Division, Kameda Medical Center, Chiba, Japan, 7 Post-Graduate Education Center, Kameda Medical Center, Chiba, Japan

* ryo.ueno@monash.edu

**Data Availability Statement:** Data cannot be shared publicly because of the confidentiality. Data are available from the Kameda Medical Centre/

## Abstract

### Background

Although machine learning-based prediction models for in-hospital cardiac arrest (IHCA) have been widely investigated, it is unknown whether a model based on vital signs alone (Vitals-Only model) can perform similarly to a model that considers both vital signs and laboratory results (Vitals+Labs model).

### Methods

All adult patients hospitalized in a tertiary care hospital in Japan between October 2011 and October 2018 were included in this study. Random forest models with/without laboratory results (Vitals+Labs model and Vitals-Only model, respectively) were trained and tested using chronologically divided datasets. Both models use patient demographics and eight-hourly vital signs collected within the previous 48 hours. The primary and secondary outcomes were the occurrence of IHCA in the next 8 and 24 hours, respectively. The area under the receiver operating characteristic curve (AUC) was used as a comparative measure. Sensitivity analyses were performed under multiple statistical assumptions.

### Results

Of 141,111 admitted patients (training data: 83,064, test data: 58,047), 338 had an IHCA (training data: 217, test data: 121) during the study period. The Vitals-Only model and Vitals+Labs model performed comparably when predicting IHCA within the next 8 hours (Vitals-Only model vs Vitals+Labs model, AUC = 0.862 [95% confidence interval (CI): 0.855–0.868] vs 0.872 [95% CI: 0.867–0.878]) and 24 hours (Vitals-Only model vs Vitals+Labs model,

Ethics Committee for researchers who meet the criteria for access to confidential data. Please contact the corresponding author and/or the medical centre ethics committee (contact via clinical_research@kameda.jp).

**Funding:** RU is supported by the Masason Foundation (MF) and has received a grant from MF. MF has not contributed to the study design, collection, management, analysis, and interpretation of data; the manuscript preparation; or the decision to submit the report for publication.

**Competing interests:** There are no conflicts of interest to declare.

AUC = 0.830 [95% CI: 0.825–0.835] vs 0.837 [95% CI: 0.830–0.844]). Both models performed similarly well on medical, surgical, and ward patient data, but did not perform well for intensive care unit patients.

## Conclusions

In this single-center study, the machine learning model predicted IHCAs with good discrimination. The addition of laboratory values to vital signs did not significantly improve its overall performance.

## Introduction

In-hospital cardiac arrests (IHCAs), which are associated with high mortality and long term morbidity, are a significant burden on patients, medical practitioners, and public health [1]. To achieve a favorable outcome, the prevention and early detection of IHCA has been proven to be essential [2,3]. Up to 80% of patients with IHCA have signs of deterioration in the eight hours before cardiac arrest [4,5], and various early warning scores based on vital signs have been developed [6–11]. The widespread implementation of electronic health records enables large datasets of laboratory results to be used in the development of early warning scores [12–15]. Recently, automated scores using machine learning models with and without laboratory results have been widely investigated, and both have achieved promising results [16–18].

However, it is unknown whether a model with vital signs alone (Vitals-Only model) performs similarly to a model that incorporates both vital signs and laboratory results (Vitals +Labs model). It is promising that prior studies that use both vital signs and laboratory results report that vital signs are more predictive than laboratory results alone for IHCA within each model [16,19]. Physiologically, changes in vital signs may be more dynamic and occur earlier than changes in laboratory results [4,5]. Computationally, the amount of vital-sign data is likely to be much larger than that of laboratory results. Vital signs are less invasive and easier to obtain, and there are many more opportunities to collect vital signs than laboratory results. For these reasons, we hypothesize that a Vitals-Only model may perform similarly to a Vitals +Labs model in the prediction of IHCA.

We were motivated to perform this study because a Vitals-Only model that performs similarly to a Vitals+Labs model is of clinical importance for the following reasons. First, a simpler model with fewer input variables can be adopted in a wide variety of settings. Vital signs are available anywhere, even in a patient's home potentially because of the development of telemetry and wearable devices, whereas laboratory results might not be available in all instances. If a model uses complex input data such as biochemical and arterial blood gas data, or complex image results, such a model may not be suitable in some hospitals, especially in low-resource settings. Second, a simple model can also be externally validated and more easily calibrated to different healthcare systems. Third, the Vitals-Only model would be non-invasive and economically feasible, because it does not require any laboratory tests, which may be physically stressful and financially burdensome for patients. Fourth, from a computational point of view, the minimal optimal model with a low data dimensionality is always better than a more complicated model as long as it has similar performance.

We hypothesized that the occurrence of an IHCA within 8 h can be predicted from vital signs alone without the need for additional laboratory tests. To assess this hypothesis, we conducted a single-center retrospective research project.

## Methods

### Study design and setting

We conducted this single-center retrospective cohort study at Kameda Medical Center, a tertiary teaching hospital in a rural area in Japan with 917 beds, which includes 16 intensive care unit (ICU) beds. This hospital has a cardiac arrest team that treats all IHCAs and consists of staff anesthetists, emergency physicians, and cardiologists. A record of the resuscitation is entered in the code blue registry immediately after an event. This hospital also has a rapid response team with an ICU senior doctor and an ICU nurse who attend calls to review patients if requested by the ward nurse. The RRT has 90–100 activations per year.

Since 1995, Kameda Medical Center has used an electronic medical record system, which collects patient information such as patient demographics, vital signs, and laboratory results. The data for this study were taken from this system.

This study was reviewed and approved by the Institutional Review Board of Kameda Medical Center (approval number: 18-004-180620). The committee waived the requirement for informed consent because of the retrospective design of the study. In addition, this study follows the Transparent Reporting of a Multivariable Prediction Model for Individual Prognosis or Diagnosis (TRIPOD) reporting guideline for prognostic studies [20].

### Study population

We included all adult patients (age ≥ 18 years old) admitted for more than 24 h between 20 October 2011 and 31 October 2018 in the study. Both ward and ICU patients were included, but emergency department patients were excluded unless they stayed in the hospital for more than 24 h. We collected the following data: demographic data on admission (i.e. age, sex, BMI, elective or emergency admission, and department of admission), eight-hourly vital signs (systolic blood pressure, diastolic blood pressure, heart rate, respiratory rate, temperature, saturation, and urinary output), and daily laboratory results (see Fig 1 for details). All the IHCA patients were identified from the code blue registry of Kameda Medical Center. Three doctors (JO, TM, and HH) retrospectively reviewed all the records and confirmed the IHCA event and the time of day it occurred.

### Prediction outcome measures

The primary outcome predicted by our model was IHCA within the next 8 h. The secondary outcome predicted by our model was IHCA within the next 24 h. All the 'expected' cardiac arrests (such as cardiac arrests in palliative care patients) without code blue responses were excluded.

### Algorithm selection

We chose a random forest model, which is a nonparametric machine learning approach that has been shown to outperform other algorithms without the need for standardization or log-transformation of the input data [16,21,22]. We chose this model for the following reasons. First, a random forest model allows us to take into account the non-linear relationships between input variables. Vital signs are known to differ among various age groups [23]. This approach enables the model to learn the age-dependent variables more precisely. Second, random forest models have relative explainability through the provision of 'feature importance'. This is a scaled measure that indicates the weighted contribution of each variable to overall prediction. It is scaled so that the most important variable is given a maximum value of 100, with less important variables given a lower value. While this allows comparison of the relative

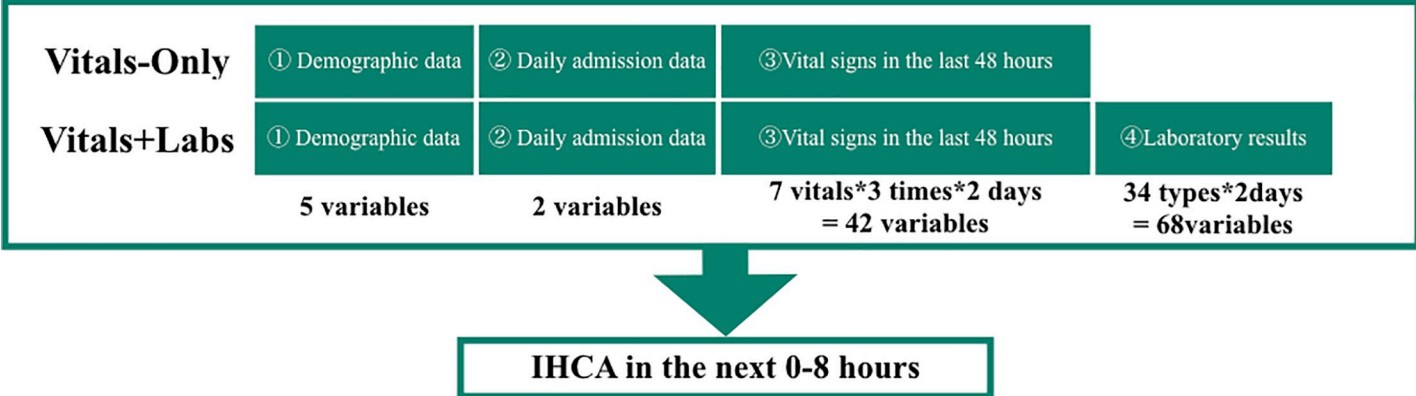

**Fig 1. Overview of the input variables for both models predicting IHCA.** Six sets of eight-hourly measured Vital signs in the last 48 hours were used in both models. Two sets of most recent Laboratory results in the last seven days were used in Lab model. Demographic data remains static whereas vital signs and laboratory results, other data, and output are tracked at regular intervals. Abbreviations: Alb, albumin; APTT, activated partial thromboplastin time; AST, aspartate aminotransferase; ALT, alanine aminotransferase; ALP, alkaline phosphatase; BMI, body mass index; BNP, brain natriuretic peptide; CK, creatine kinase; CKMB, creatine kinase–muscle/brain; Cre, creatinine; CRP, C-reactive protein; eGFR, estimated estimated Glomerular. Filtration Rate; GGT, gamma-glutamyl transferase; Glu, glucose; Hb, haemoglobin; Hct, haematocrit; IHCA, in-hospital cardiac arrest; LD, lactate dehydrogenase; MCH, mean corpuscular hemoglobin; MCHC, mean corpuscular haemoglobin concentration; MCV, mean corpuscular volume; Plt, platelets; PT, prothrombin time; PT-INR, prothrombin time international normalized ratio; RBC, red blood cells; TP, total protein; T-Bil, total bilirubin; WBC, white blood cells.

importance of each variable within a model, it does not allow comparison of the relative contribution of the same variables between different models.

## Statistical methods

Patient data were divided into training data and test data according to admission date (training data dates: 20 October 2011 to 31 December 2015; test data dates: 1 January 2016 to 31 October 2018) [17]. As summarized in Fig 1, predictions were made every 8 h using patient demographics and eight-hourly vital signs in the last 48 h in the Vitals-Only model. The Vitals+Labs model was the same as the Vitals-Only model but with the addition of two sets of laboratory results obtained in the last seven days, as previously reported [17,24]. As shown in Fig 2, an ensemble model of trees was developed using training data after bagging, and then the model was validated using the test dataset. (See the S1 File for further details.)

We measured the prediction performance of each model by computing the following values: (1) the C statistic (i.e. the area under the receiver operating characteristic [ROC] curve); (2) the prospective prediction results (i.e. sensitivity, specificity, positive predictive value, negative predictive value, positive likelihood ratio, and negative likelihood ratio) at Youden's index;

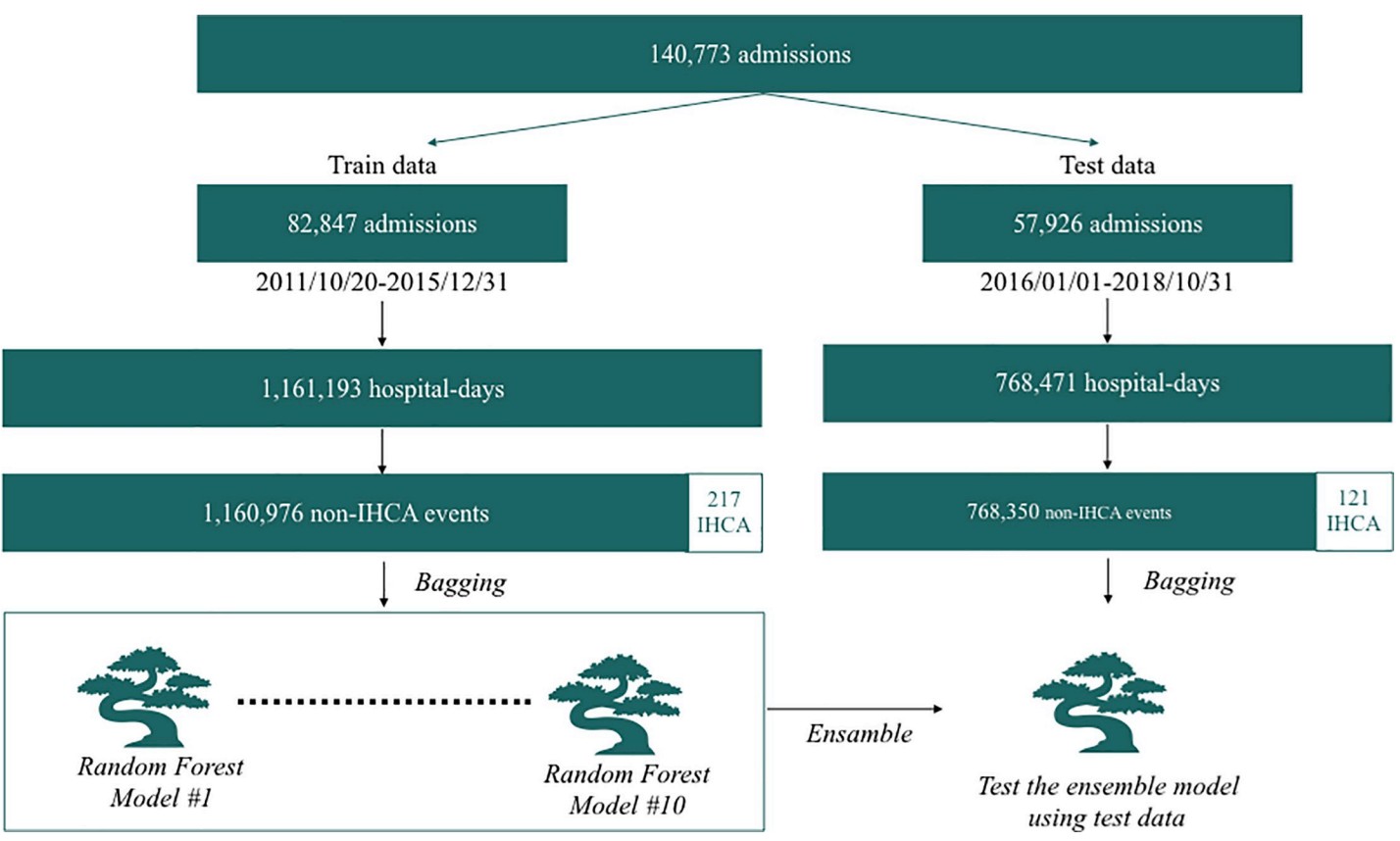

**Fig 2. Architectural overview of data extraction and representation.**

and (3) the calibration curve. To gain insight into the contribution of each predictor to our model, we calculated their feature importance with respect to the primary outcome.

Sensitivity analyses were planned a priori under various statistical assumptions. To compare the Vitals-Only model with the Vitals+Labs model in various populations, we tested the performance of our models in patient subsets (i.e. medical admission or surgical admission; ward admission or ICU admission). To assess the impact of missing data, we repeated the primary analysis with different imputation methods (see the S1 File for further details).

Values that are clearly errors (e.g. systolic blood pressure >300 mmHg) were removed, as described in [17,25] (see the S1 File for further details). No other preprocessing (e.g. normalization or log-transformation) of the dataset was performed. Missing values were imputed with the patient's last measured value for that feature or the median value of the entire sample if a patient had no previous values, as described in [14,15,26]. If more than 50% of data for a particular vital sign or laboratory result was missing in the entire dataset, the feature was converted to a binary value (1 denotes a measured value and 0 denotes a missing value) [27]. Various types of missing imputations were performed as part of the sensitivity analyses (see the S1 File for further details). Categorical variables were expressed as percentages, whereas continuous variables were described as means (± standard deviations, SDs) or median (± interquartile ranges, IQRs). The analyses were performed using Python 3.7.0, and we have made the analysis code publicly available (https://github.com/liyuan9988/Automet).

## Results

### Details of the patient cohort and IHCAs

A total of 143,190 admissions of adult patients were recorded during the study period. After 2,069 admissions less than 24 h in length were excluded, the remaining 141,111 admissions were used in the analysis. Among these admissions, 338 IHCAs were recorded. Patients characteristics were similar in both the training and test data (Table 1). The percentage of missing data for each variable is summarized in S1 Table in S1 File; 11/130 (8.5%) variables with >50% missing values were converted into binary values following the rule described in the Method section. As summarized in S2 Table in S1 File, patients who suffered an IHCA had characteristics different from those who did not. Of note, IHCA patients were older, frequently male, and admitted with non-surgical conditions. Of the IHCA patients, almost 40% were admitted to the Cardiology department. The next most common admission departments were Hematology and General Internal Medicine, each accounting for approximately 10% of the IHCA patients.

### Predictive ability of the Vitals-Only model and Vitals+Labs model

As summarized in Table 2, the Vitals-Only model (AUC = 0.862 [95% confidence interval (CI): 0.855–0.868]) and Vitals+Labs model (0.872 [95% CI: 0.867–0.878]) had similar performance in predicting the occurrence of an IHCA in the next 8 h. In addition, similar results were obtained for the prediction of IHCA in the next 24 h (Vitals-Only model vs Vitals+Labs model, AUC = 0.830 [95% CI: 0.825–0.835] vs 0.837 [95% CI: 0.830–0.844]). At Youden's index, both models achieved similar prospective prediction results for both positive and negative prediction values (Vitals-Only model vs Vitals+Labs model, positive prediction value = 0.035 [95% CI: 0.029–0.041] vs 0.044 [95% CI: 0.035–0.053], negative prediction value = 0.998 [95% CI: 0.997–0.998] vs 0.997 [95% CI: 0.997–0.998]).

### Calibration plot

As shown in Fig 3, the calibration curve of both models were similarly far from the diagonal of the calibration plot (Vitals-Only model vs Vitals+Labs model, Hosmen-Lemeshow C-statistics = 8592.40 vs 9514.76, respectively). For the highest risk group, the occurrence of IHCA was 20%–30% in both models.

### Sensitivity analysis with different models

We performed several sensitivity analyses under various statistical assumptions. Our results were unchanged when we applied the models to medical patients (Vitals-Only model vs Vitals+Labs model, AUC = 0.869 [95% CI: 0.862–0.877] vs 0.876 [95% CI: 0.870–0.882]) or surgical patients (Vitals-Only model vs Vitals+Labs model, AUC = 0.806 [95% CI: 0.797–0.816] vs 0.825 [95% CI: 0.802–0.848]). Likewise, the Vitals-Only model was not inferior to the Vitals+Labs model among ward patients (Vitals-Only model vs Vitals+Labs model, AUC = 0.879 [95% CI: 0.871–0.886] vs 0.866 [95% CI: 0.858–0.874]).

However, the discrimination of both model types was poor when applied to the ICU population. The Vitals+Labs model outperformed Vitals-Only model for ICU patients (Vitals-Only model vs Vitals+Labs model, AUC = 0.580 [95% CI: 0.571–0.590] vs 0.648 [95% CI: 0.635–0.661]; Table 3). Finally, our results were similar regardless of the type of imputation. The Vitals-Only model had a performance similar to that of the Vitals+Labs model with AUCs ranging between 0.80 and 0.90 for all four imputation methods (S3 Table in S1 File).

**Table 1. Characteristics of study population.**

| Comparison of Training/Test data | | |
|---|---|---|
| | Training | Test |
| Study Period | Oct 2011- Dec 2015 | Jan 2016—Oct 2018 |
| Total Admissions, n | 83,064 | 58,047 |
| Age, y, median (IQR) | 64 (54; 77) | 65 (56; 77) |
| Male sex, n (%) | 41,868 (50) | 29,210 (50) |
| BMI, kg/m$^2$, median (IQR) | 22.9 (20.5; 25.4) | 23.1 (20.7; 25.7) |
| Surgical Patients, n (%) | 41,337 (50) | 29,491 (51) |
| Emergency Admission, n (%) | 17,853 (21) | 14,673 (25) |
| Patient with IHCA, n (%) | 217 (0.3) | 121 (0.2) |
| Patient with In-Hospital Death, n (%) | 2,577 (3.1) | 1,729 (3.0) |

| Comparison of Patients with/without IHCA | | | | |
|---|---|---|---|---|
| | Training | | Test | |
| | IHCA | non-IHCA | IHCA | non-IHCA |
| Total, n | 217 | 82847 | 121 | 57926 |
| Male, n (%) | 142 (65.4) | 41726 (50.4) | 77 (63.6) | 29133 (50.3) |
| Age, median (IQR) | 75 (66:81) | 67 (54:77) | 72 (63:81) | 68 (56:77) |
| BMI, median (IQR) | 24 (21;26) | 23 (21;25) | 24 (20;28) | 23 (21;26) |
| Emergency admission, n (%) | 63 (29.0) | 17790 (21.5) | 27 (22.3) | 14646 (25.3) |
| Surgical admission, n (%) | 50 (23.0) | 41287 (49.8) | 24 (19.8) | 29467 (50.9) |
| History of previous IHCA in the same admission, n (%) | 26 (12.0) | 15 (0.0) | 14 (11.6) | 11 (0.0) |
| In-hospital death, n (%) | 164 (75.6) | 2413 (2.9) | 88 (72.7) | 1641 (2.8) |
| Primary Admission Department, n (%) | | | | |
| Breast Surgery | 0 ( 0.0) | 3592 (4.3) | 0 (0.0) | 2710 (4.7) |
| Cardiology | 96 (44.2) | 6556 (7.9) | 46 (38.0) | 5299 (9.1) |
| Cardiovascular Surgery | 14 ( 6.5) | 1412 (1.7) | 4 (3.3) | 768 (1.3) |
| Dermatology | 0 ( 0.0) | 136 (0.2) | 0 (0.0) | 68 (0.1) |
| Emergency Medicine | 2 ( 0.9) | 1758 (2.1) | 6 (5.0) | 709 (1.2) |
| Endocrinology | 0 (0.0) | 385 (0.5) | 0 (0.0) | 307 (0.5) |
| ENT | 3 (1.4) | 1594 (1.9) | 1 (0.8) | 1287 (2.2) |
| Gastroenterology | 11 (5.1) | 11282 (13.6) | 4 (3.3) | 6082 (10.5) |
| General Internal Medicine | 13 (6.0) | 3758 (4.5) | 15 (12.4) | 3902 (6.7) |
| General Surgery | 13 (6.0) | 6161 (7.4) | 4 (3.3) | 4387 (7.6) |
| Hematology | 14 (6.5) | 1850 (2.2) | 13 10.7) | 1594 (2.8) |
| Infectious Disease | 0 (0.0) | 47 (0.1) | 0 (0.0) | 91 (0.2) |
| Nephrology | 14 (6.5) | 2418 (2.9) | 4 (3.3) | 1449 (2.5) |
| Neurology | 3 (1.4) | 2747 (3.3) | 5 (4.1) | 1668 (2.9) |
| Neurosurgery | 1 (0.5) | 1812 (2.2) | 2 (1.7) | 1145 (2.0) |
| Obstetrics and Gynecology | 3 (1.4) | 9080 (11.0) | 0 (0.0) | 5107 (8.8) |
| Oncology | 9 (4.1) | 3387 (4.1) | 4 (3.3) | 1447 (2.5) |
| Opthalmology | 0 (0.0) | 2206 (2.7) | 0 (0.0) | 1562 (2.7) |
| Oral and Maxillofacial Surgery | 0 (0.0) | 1110 (1.3) | 2 (1.7) | 648 (1.1) |
| Orthopedics | 6 (2.8) | 3070 (3.7) | 3 (2.5) | 2312 (4.0) |
| Palliative Medicine | 0 (0.0) | 3 (0.0) | 0 (0.0) | 1 (0.0) |
| Pediatrics | 0 (0.0) | 3 (0.0) | 0 (0.0) | 2 (0.0) |
| Plastic Surgery | 2 (0.9) | 1052 (1.3) | 0 (0.0) | 814 (1.4) |
| Psychiatry | 0 (0.0) | 732 (0.9) | 0 (0.0) | 390 (0.7) |
| Pulmonology | 6 (2.8) | 4868 (5.9) | 4 (3.3) | 3984 (6.9) |

(*Continued*)

**Table 1.** (Continued)

| | | | | |
|---|---|---|---|---|
| Rehabilitation | 0 (0.0) | 1306 (1.6) | 0 (0.0) | 844 (1.5) |
| Rheumatology | 1 (0.5) | 1936 (2.3) | 0 (0.0) | 1055 (1.8) |
| Spinal Surgery | 1 (0.5) | 2082 (2.5) | 0 (0.0) | 1354 (2.3) |
| Sports medicine | 0 (0.0) | 720 (0.9) | 1 (0.8) | 675 (1.2) |
| Thoracic Surgery | 2 (0.9) | 1135 (1.4) | 2 (1.7) | 1201 (2.1) |
| Urology | 3 (1.4) | 4649 (5.6) | 1 (0.8) | 5064 (8.7) |

Data are n (%) or median (IQR)

## Discussion

### Summary of key findings

In this retrospective study with 141,111 admissions, we compared two prediction models for IHCA, one using vital signs and patient background only and one using the same information plus laboratory results. The Vitals-Only model yielded a performance that was similar to that of Vitals+Labs model for all data except for that of ICU patients, where discrimination was poor for both models.

### Relationship with prior literature

Prior studies have extensively investigated the importance of early prediction of IHCA [2–5]. It is known that monitored or witnessed IHCAs have more favorable outcomes even after IHCA compared to events that are unmonitored or unwitnessed [2,3]. To support the necessity of preventive monitoring, prior studies found that clinical deterioration is common prior to cardiac arrest [4,5]. Based on these findings, various early warning scores have been developed, ranging from an analog model based on vital signs to a digital scoring system using both vital signs and laboratory results [6–15]. Recently, a variety of early warning scores have utilized machine learning to account for the non-linear relationships in various input variables [16–18].

Various studies on models that use both vital signs and laboratory results have reported that vital signs are more predictive than laboratory results for IHCA. In a study by Churpek and colleagues, the top five most predictive variables for the composite outcome including IHCA were all vital signs [16]. A similar result was obtained in [19]. We note that our study has similar results to the results of those studies in that higher weights are given to heart rate, blood pressure, and age in our model. However, our model was unable to learn the importance of respiratory rate because of the high rate of missing respiratory rate data in our dataset (Fig 4). Similar to our study, a few studies have investigated the change in predictive performance using different sets of input features. Churpek and colleagues reported a prediction model for IHCA using both vital signs and laboratory results [15]. Their model performed better than

**Table 2. Predictive performance of each model for the in-hospital cardiac arrest in the next 0–8 hours.**

| Model | AUC (95%CI) | PPV (95%CI) | NPV (95%CI) | Sensitivity (95%CI) | Specificity (95%CI) | PLR (95%CI) | NLR (95%CI) |
|---|---|---|---|---|---|---|---|
| Vitals-Only model | 0.862 [0.855; 0.868] | 0.035 [0.029; 0.041] | 0.998 [0.997; 0.998] | 0.817 [0.754; 0.880] | 0.772 [0.716; 0.827] | 3.58 [2.92; 4.25] | 0.238 [0.172; 0.304] |
| Vitals+Labs model | 0.872 [0.867; 0.878] | 0.044 [0.035; 0.053] | 0.997 [0.997; 0.998] | 0.770 [0.719; 0.821] | 0.830 [0.781; 0.879] | 4.52 [3.56; 5.50] | 0.277 [0.229; 0.325] |

Abbreviations: 95%CI, 95% confidence interval; AUC, area under the receiver operating characteristic curve; PPV, positive predictive value; NPV, negative predictive value; PLR, positive likelihood ratio; NRL, negative likelihood ratio

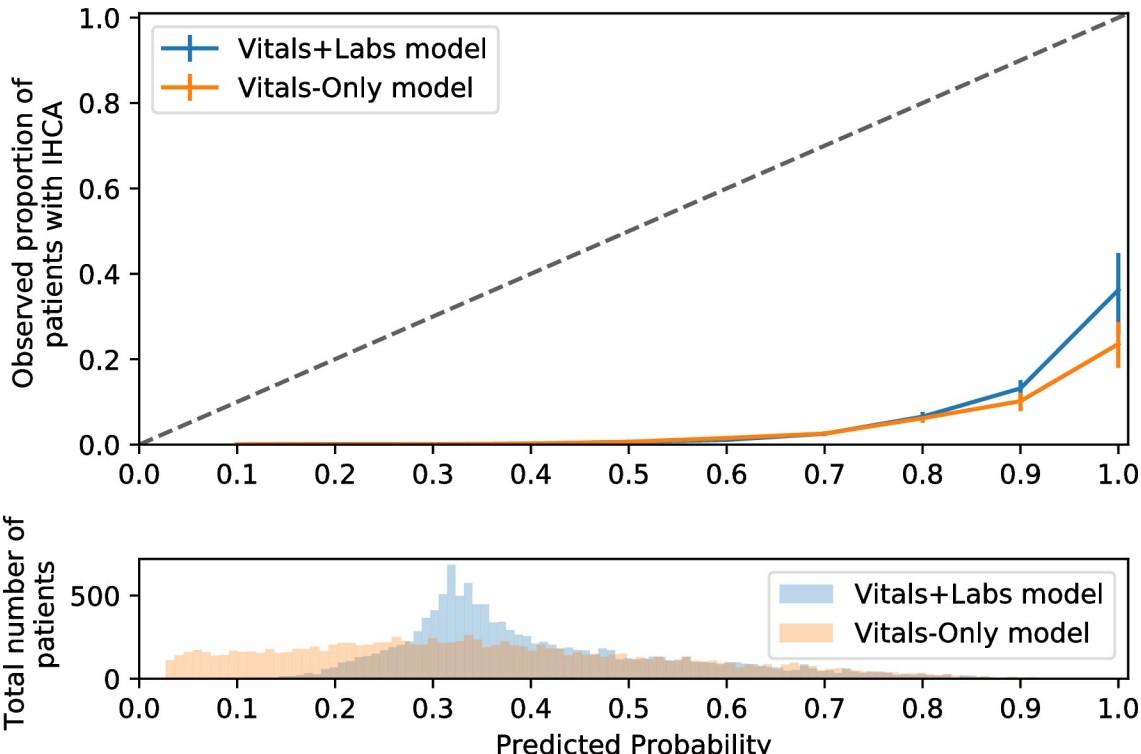

**Fig 3. Calibration plot.** The x-axis summarizes the predicted probability of having a cardiac arrest in the next 8 hours, whereas the y-axis shows the observed proportions of patients who have a cardiac arrest in next 8 hours. Histogram below the calibration plot summarizes the distribution of predicted probability amongst all the patients in our dataset. Abbreviations: IHCA; inhospital cardiac arrest.

their previously published model, which was only based on vital signs [9]. However, those models were developed and validated in different cohorts with different methodologies. In a short communication by Kellett and colleagues, a score based solely on vital signs was more

**Table 3. Predictive performance of each model in sets of sensitivity analyses.**

| | AUC (95%CI) | PPV (95%CI) | NPV (95%CI) | Sensitivity (95%CI) | Specificity (95%CI) | PLR (95%CI) | NLR (95%CI) |
|---|---|---|---|---|---|---|---|
| **Medical** | | | | | | | |
| Vitals-Only | 0.869 [0.862; 0.877] | 0.047 [0.034; 0.060] | 0.997 [0.996; 0.998] | 0.803 [0.714; 0.892] | 0.783 [0.701; 0.865] | 3.70 [2.63; 4.79] | 0.251 [0.160; 0.342] |
| Vitals+Labs | 0.876 [0.870; 0.882] | 0.052 [0.039; 0.066] | 0.997 [0.996; 0.998] | 0.796 [0.733; 0.859] | 0.809 [0.746; 0.871] | 4.16 [3.01; 5.34] | 0.253 [0.193; 0.313] |
| **Surgical** | | | | | | | |
| Vitals-Only | 0.806 [0.797; 0.816] | 0.020 [0.016; 0.025] | 0.998 [0.998; 0.998] | 0.708 [0.661; 0.754] | 0.819 [0.768; 0.871] | 3.91 [3.00; 4.82] | 0.357 [0.316; 0.398] |
| Vitals+Labs | 0.825 [0.802; 0.848] | 0.018 [0.011; 0.026] | 0.998 [0.998; 0.999] | 0.707 [0.604; 0.811] | 0.795 [0.711; 0.879] | 3.45 [2.06; 4.87] | 0.368 [0.278; 0.458] |
| **ICU** | | | | | | | |
| Vitals-Only | 0.580 [0.571; 0.590] | 0.346 [0.338; 0.353] | 0.905 [0.881; 0.930] | 0.889 [0.848; 0.930] | 0.384 [0.342; 0.426] | 1.44 [1.39; 1.49] | 0.288 [0.209; 0.372] |
| Vitals+Labs | 0.648 [0.635; 0.661] | 0.382 [0.331; 0.434] | 0.859 [0.775; 0.944] | 0.725 [0.472; 0.977] | 0.560 [0.300; 0.821] | 1.65 [1.32; 2.04] | 0.492 [0.178; 0.874] |
| **Ward** | | | | | | | |
| Vitals-Only | 0.879 [0.871; 0.886] | 0.022 [0.018; 0.026] | 0.999 [0.998; 0.999] | 0.81 [0.752; 0.868] | 0.791 [0.739; 0.844] | 3.88 [3.13; 4.63] | 0.240 [0.180; 0.300] |
| Vitals+Labs | 0.866 [0.858; 0.874] | 0.024 [0.017; 0.032] | 0.998 [0.998; 0.999] | 0.782 [0.705; 0.859] | 0.814 [0.738; 0.890] | 4.21 [2.85; 5.59] | 0.268 [0.196; 0.340] |

Abbreviations: 95%CI, 95% confidence interval; AUC, area under the receiver operating characteristic curve; PPV, positive predictive value; NPV, negative predictive value; PLR, positive likelihood ratio; NLR, negative likelihood ratio

## Vitals-Only model

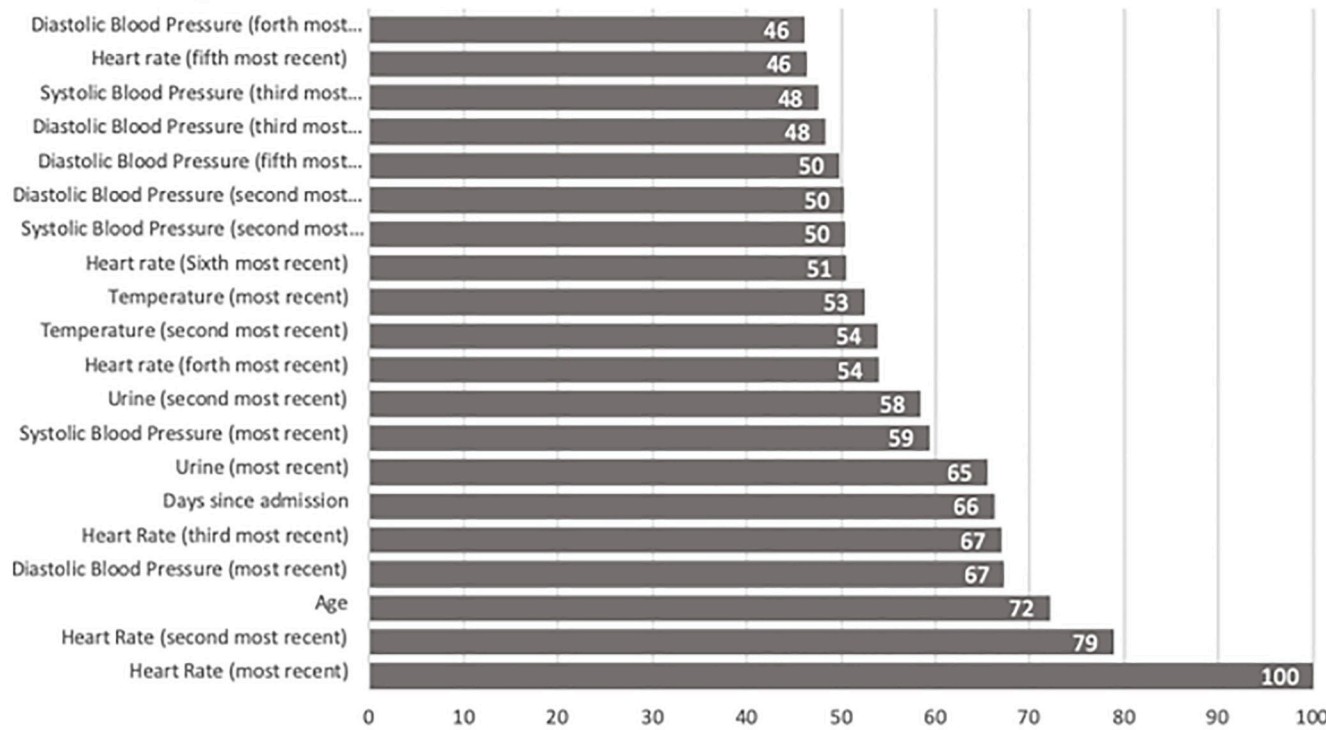

## Vitals+Labs model

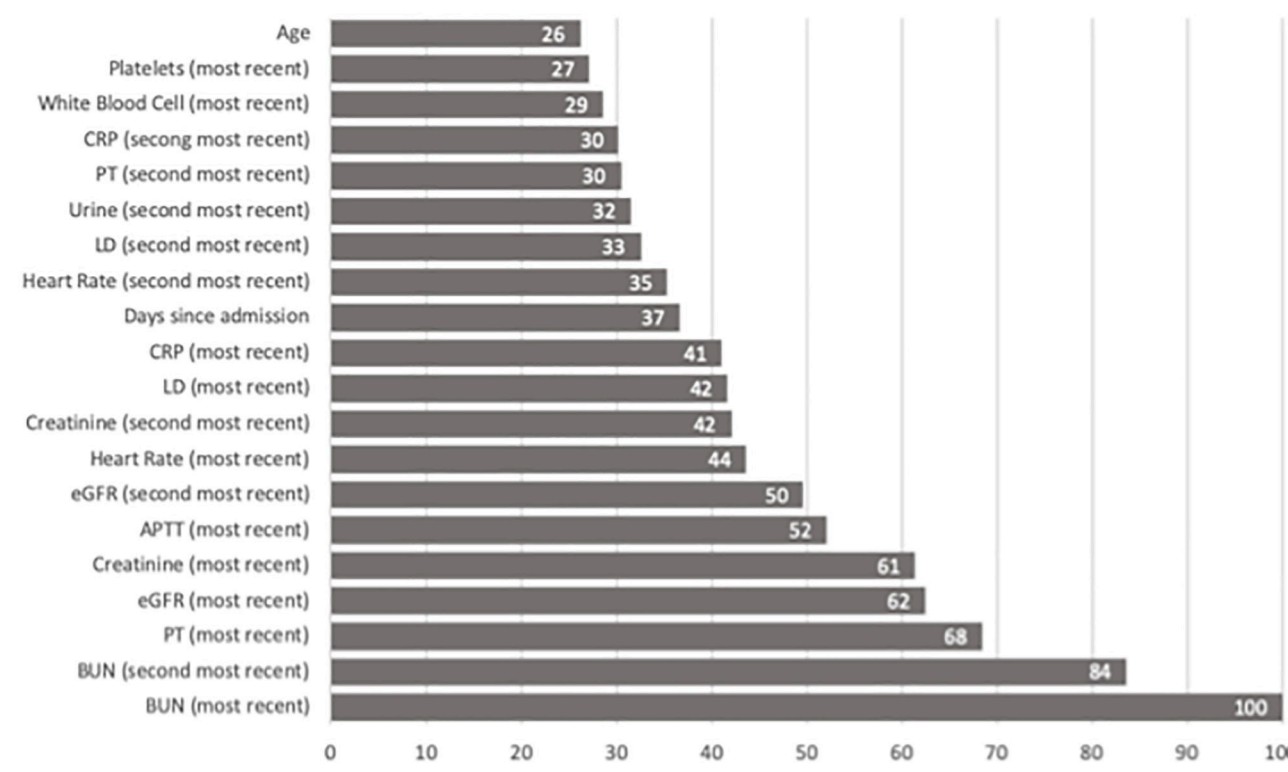

**Fig 4. Feature importance in predicting subsequent occurrence of IHCA.** Importance of each predictor in the 2 different random forest models: Vitals-Only model and Vitals+Labs model. 20 most important variables were summarized. Abbreviations: APTT, activated partial thromboplastin time; BUN, blood urea nitrogen; CRP, C-reactive protein; LD, lactate dehydrogenase; PT, prothrombin time.

predictive than other scores using both vital signs and laboratory results to predict in-hospital mortality [28]. To date, the best performing model for IHCA (AUC = 0.85) was built solely using vital signs by Kwon and colleagues. The authors argued that the addition of laboratory results should improve performance, but this has not yet been confirmed [17].

## Interpretation of the results

In our dataset, both the Vitals-Only and Vitals+Labs models were capable of predicting early IHCA (within the next 8 h) and late IHCA (within the next 24 h). There are three possible reasons for this result. First, physiologically, eight-hourly measured vital signs obtained in the last 48 h might reflect acute deterioration more sensitively than laboratory results obtained within the last seven days. Second, doctors and nurses might have intervened to treat patients with abnormal laboratory results to prevent physiological deterioration. As a result, such deterioration might not have resulted in IHCA [29]; this could be one reason why abnormal laboratory results did not directly lead to IHCA. However, our data lacks this treatment information, so we are unable to verify this supposition conclusively. Third, our model might have failed to learn from the laboratory results. However, laboratory results, which have been clinically and theoretically proven to be associated with IHCA [13,15,30], have a high feature importance in the model (Fig 4), which indicates they were successfully learned. In addition, the newer variables (i.e. the variable measured closer to the time of prediction) of both vital signs and laboratory results have heavier weights in the models than older variables, which also evidences successful feature learning.

Regardless of whether the admission was classified as medical or surgical, we did not observe a notable difference in the performances of the Vitals-Only model and Vitals+Labs model. This result is consistent with a prior study that compares the performance of the National Early Warning Score among medical and surgical patient populations [31]. However, our model did not predict IHCA well in ICU populations. We believe this is a result of the continuous vital-sign monitoring and more frequent interventions in ICU. Even with the same abnormal vital signs, ICU patients are more likely to receive clinical interventions to prevent IHCA than are ward patients. Given the heterogeneity of patient backgrounds, interventions, and the amount of available data, a tailored prediction model should be developed specifically for ICU patients in future trials.

## Strengths

Our study has several strengths. First, our model aims to predict unexpected IHCAs rather than a combined outcome or surrogate outcome. Unlike other outcomes such as ICU transfer, unexpected IHCA is always objective and always necessitates clinical intervention.

Second, to obtain a straightforward interpretation of the prediction, we used a classification model rather than a time-to-event model. Classification models were shown to outperform time-to-event models in predicting hospital deterioration in [22].

Third, all imputation methods assessed in this study are prospectively implementable. A prior study showed that the timing of each laboratory test itself has predictive value [27]. Our model used this information for variables with >50% of the data missing, thereby maximizing the utilization of the available data.

## Limitations and future work

There are various limitations in our study. First, this study was conducted in a single center. Our results might be biased by the patient backgrounds and clinical practice of a tertiary care center. If the frequency or methodology of vital measurements or laboratory tests varies in other hospitals, their predictive values might be different. Even though our model shares many of the characteristics identified in other clinical deteriorations of patients in prior studies [15,19], further calibration and validation of our results in different clinical settings are warranted.

Second, we were unable to obtain any data regarding treatment. Theoretically, the patient management (e.g. treatment practice and staffing levels) could have changed over the study period. Such information might have improved the model performance, but we were unable to obtain it.

Third, some of our input features were missing. For example, various studies have stressed the importance of respiratory rate, which is mostly missing in our dataset and hence was converted into binary values. However, data will inevitably be missing when this model is used in a real-world setting. In addition, our results were found to be robust after attempting various imputation methods; we hence believe our approach is appropriate for developing and validating a model for bedside use.

Fourth, a low positive predictive value is a global issue in all prediction models of rare events such as IHCA (0.4% of prevalence) [25,32]. Though no false positives might be better than many false negatives with respect to cardiac arrest, each hospital needs to seek its own balance to optimize the tradeoff between false alarms and overlooking IHCA patients. In this study, we focused on a *comparison* of the Vitals-Only model and Vitals+Labs model at a widely investigated threshold (i.e. Youden's index), rather than the *provision* of a highly predictive model. A different threshold and more sophisticated feature engineering might aid in developing a model with a higher positive predictive value. In addition, it would be worth investigating the outcomes of 'false positive' patients in future studies. These are patients flagged as being at risk of cardiac arrest, and false positives could still be identified as 'at risk' patients, facilitating early intervention such as a rapid response team.

Fifth, we used eight-hourly discrete vital signs rather than a continuous dataset. This approach reflects the clinical practice in our hospital (vital signs are measured three times a day in most of our patients) and prior studies [15,33]. Sometimes, the timing and frequency of vital-sign measurement reflects the concern of medical professionals [34]. Hence, the addition of such information might improve the performance of the model.

Sixth, the aim of our model is discrimination rather than calibration. That is, the aim of this research is to *compare* the discrimination performance of two models rather than to *provide* a well-calibrated risk score. However, if we were to implement the Vitals-Only model in an actual clinical setting, a well-calibrated model might be highly valuable for clinical decision-making because it enables clinicians to interpret the predicted probability as a risk score. Although various clinical tests still solely focus on discrimination (e.g. pregnancy tests and fecal occult blood tests), they should ideally be all calibrated. As shown in Fig 3, both models were similarly far from the diagonal of the calibration curve. It is promising that the highest risk group had a 20%–30% occurrence of IHCA, but future studies should investigate the more sophisticated calibration of the Vitals-Only model before its clinical application.

Seventh, the random forest model is only partially interpretable. The random forest model provides the feature importance, which shows the weight of each variable in the model and enables us to assess the clinical validity of the model's decision process. However, the random forest model does not provide the reason why a patient was flagged as likely to have an IHCA

in the next 24 h. If a clinician were keen to obtain such information, our model would remain a 'black box'. In addition, different clinicians could have different interpretations for the flag if this model were unable to provide any reason for the alert. Despite these drawbacks, we used the random forest model for the performance reasons summarized in the Methods. Moreover, for an event such as an IHCA, a clinician's response to an alert may be quite simple. Regardless of its cause, a flag will necessitate a clinical review, and clinicians will be able to synthesize all the available information for clinical decision-making. In such a process, the most important part of the alert is the act of alerting the clinician rather than providing a reason for the alert. Moreover, some cardinal investigations in medicine are 'black boxes' in nature. Not all clinicians understand why a certain saddle shape in an electrocardiogram is highly associated with a fetal congenital arrhythmia, but all emergency physicians rely on this waveform in their daily practice. Ultimately, the tradeoff between a model's performance and its interpretability will depend on medical professionals at the bedside. In future study, such factors will be highly important to consider before the Vitals-Only model can be applied in clinical practice.

Eighth, we did not provide any clinical practice as a benchmark comparison for the Vitals-Only model. This study focuses on the comparison of Vitals-Only model and Vitals+Labs model rather than providing a single model for clinical use. However, it will be essential to provide appropriate benchmarks in order to assess a model's clinical utility.

Finally, as is often the case with all prediction models, we do not yet know whether our prediction model would actually improve the trajectory of patients at risk. Clinically, we have found little evidence of the reliability, validity, and utility of these systems. However, our results are persuasive enough to facilitate a prospective validation of the Vitals-Only model.

## Implications of this research

While it is important to achieve the best performance by using all the data available, it is also important to focus on developing a simple model for better generalizability [35,36]. We hope our results will stimulate further investigations into and implementations of such a model.

## Conclusions

In this single-center retrospective cohort study, the addition of laboratory results to a patient's vital signs did not increase the performance of a machine-learning-based model for predicting IHCA. The prediction of IHCAs for patients in the ICU was found to be unreliable. However, the simpler Vitals-Only model performed well enough on other patient types to merit further investigation.

## Supporting information

**S1 File.**
(DOCX)

## Acknowledgments

We sincerely appreciate Mr Hiroshi Harada, Mr Masahiko Sato, and all the other members of the Data Science Team at Kameda Medical Center for their tremendous help in data collection.

## Notation of prior abstract publication/presentation

The preliminary results of this study were presented at the International Society for Rapid Response Systems 2019 Annual General Meeting in Singapore and World Congress of Intensive Care in Melbourne.

## Author Contributions

**Conceptualization:** Ryo Ueno, Wataru Uegami, Hiroki Matsui, Jun Okui, Hiroshi Hayashi, Toru Miyajima, David Pilcher, Daryl Jones.

**Data curation:** Ryo Ueno, Wataru Uegami, Jun Okui, Hiroshi Hayashi, Toru Miyajima.

**Formal analysis:** Ryo Ueno, Liyuan Xu.

**Funding acquisition:** Ryo Ueno.

**Investigation:** Ryo Ueno.

**Methodology:** Ryo Ueno, Liyuan Xu, Wataru Uegami, Hiroki Matsui, David Pilcher, Daryl Jones.

**Project administration:** Ryo Ueno, Wataru Uegami, Jun Okui, Hiroshi Hayashi, Toru Miyajima, Yoshiro Hayashi.

**Resources:** Ryo Ueno.

**Software:** Ryo Ueno, Liyuan Xu.

**Supervision:** David Pilcher, Daryl Jones.

**Visualization:** Ryo Ueno, Liyuan Xu.

**Writing – original draft:** Ryo Ueno, David Pilcher, Daryl Jones.

**Writing – review & editing:** Ryo Ueno, Liyuan Xu, Hiroki Matsui, Yoshiro Hayashi, David Pilcher, Daryl Jones.

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
