## [Decision Letter · Decision Letter 0]

28 Feb 2020

PONE-D-20-03952

Additional Value of Laboratory Results over Vital Signs in a Machine Learning Algorithm to Predict In-Hospital Cardiac Arrest: A Single-Centre Retrospective Cohort Study

PLOS ONE

Dear Ueno Ryo,

Thank you for submitting your manuscript to PLOS ONE. After careful consideration, we feel that it has merit but does not fully meet PLOS ONE’s publication criteria as it currently stands. Therefore, we invite you to submit a revised version of the manuscript that addresses the points raised during the review process.

ACADEMIC EDITOR: The reviewers have raised a number of points which we believe major modifications are necessary to improve the manuscript, taking into account the reviewers' remarks. Our expert reviewers, especially statistician review has concerns on methods and statistical analysis, and Tuhe use of random forest does not lead to a generalizable model. Please consider and address each of the comments raised by the reviewers before resubmitting the manuscript. This letter should not be construed as implying acceptance, as a revised version will be subject to re-review.

We would appreciate receiving your revised manuscript by Apr 13 2020 11:59PM. To enhance the reproducibility of your results, we recommend that if applicable you deposit your laboratory protocols in protocols.io, where a protocol can be assigned its own identifier (DOI) such that it can be cited independently in the future. For instructions see: http://journals.plos.org/plosone/s/submission-guidelines#loc-laboratory-protocols

We look forward to receiving your revised manuscript.

Kind regards,

Wisit Cheungpasitporn, MD, FACP

Academic Editor

PLOS ONE

Journal Requirements:

1. Thank you for including your competing interests statement; "No"

3. Please include your tables as part of your main manuscript and remove the individual files. Please note that supplementary tables (should remain/ be uploaded) as separate "supporting information" files

Reviewers' comments:

Reviewer's Responses to Questions

**Comments to the Author**

1. Is the manuscript technically sound, and do the data support the conclusions?

Reviewer #1: Partly

Reviewer #2: Partly

2. Has the statistical analysis been performed appropriately and rigorously? 

Reviewer #1: No

Reviewer #2: No

3. Have the authors made all data underlying the findings in their manuscript fully available?

Reviewer #1: Yes

Reviewer #2: Yes

4. Is the manuscript presented in an intelligible fashion and written in standard English?

Reviewer #1: No

Reviewer #2: Yes

5. Review Comments to the Author

Reviewer #1: The topic of this article is important however substantial revisions are suggested to improve the quality and clarify of the work. Overall the language has to be improved. Additionally, authors have often used uncommon words such as “parsimonious” , “ plausible” , etc — it is suggested to use simpler words and sentences. Although the meaning remains the same , but a simple sentence/word is less likely to be misinterpreted and is also well received by international readers.

1. Introduction: it is too short and needs major changes. The research gap is not well supported. It is suggested to clarify the state of the art, the motivation, and the research question.

2. Method: the subsection “outcome” might confuse readers. Please assign a different subsection name. Also provide with a reasoning for the choice of your ML model. It is also suggested to list the limitations of the selected model and address them in the paper.

3. Discussion: it required major rework. Not only English but also the flow has to be improved. Many claims needs citations. I have highlighted some of them :

“Prior studies have extensively investigated the importance of early prediction of IHCA.” [cite]

“Clinical deterioration is common prior to the cardiac arrest [4,5] and monitored or witnessed patient have more favorable outcomes [2,3] “ — this sentence is very confusing. I suggest rephrasing it.

“ Historically, multiple early warning scores have been developed, ranging from an analogue model based on vital signs to a digital scoring system using both vital signs and laboratory resu...” [cite]

“ Similar results were obtained in other studies [28]. ” — cite more studies or rephrase the sentence (studies — study) .

In the discussion I also suggest to include subsections discussing the effects biases in such ML models and how this has been addressed in the study. What are the risks associated?. Is the outcome clinically meaningful? (Please read the following to build upon discussion section: https://doi.org/10.12968/bjhc.2019.0066 ; https://doi.org/10.1126%2Fscience.aaw0029)

Also a section dedicated to future research direction is highly recommended.

Reviewer #2: General Comment: The authors present their findings from a diagnostic model development project, aiming to compare two machine learning algorithms for predicting in-hospital cardiac arrest: a standard approach using both vital sign and laboratory measures from an EHR, and a reduced model using only the vital sign components. They report that both models provide similar discrimination in-hospital cardiac arrest occurrence, with similar results in several settings. The debilitating omission in this manuscript is that authors provide only the crudest of calibration/validation measurements, and double down on their refusal to calibrate in their Discussion. The use of random forest does not lead to a generalizable model, at least how used here.

Specific Comments:

1. The Introduction and Methods section combined are shorter than the Discussion. As a result there is little motivation of the project and the methods used by the authors are not reported in enough detail, particularly the analytic approach (more on both of these below).

2. Introduction, Second paragraph: The authors main justification of why they would like to investigate omitting laboratory measurements is that the effect of doing so is unknown. While true, this is hardly justification for why they should feel doing so will not affect the diagnostic tool. There has to be a conceptual reason as to why they think this would work, and that reason(s) needs to be reported.

3. Methods Section, Study Population: The authors report that the vital signs and laboratory results are shown in detail in Figure 1. However, this figure shows nothing of the sort.

4. Methods Section, Statistical Methods: The authors use the random forest model for the prognostic tool since -- as they state -- it usually outperforms other algorithms. The reason for that is because it leads to models of great complexity, and model-averages across numerous well-fitting models. This leads to an uninterpretable model, which limits the ability for anyone (let alone diagnostic clinicians or scientists) who might want to use it. Because of this method, the authors do not -- and could not even if they wanted to -- report any prognostic model. As such, there approach is not reproducible and appears of little value then "use random forests, add these vital sign measurements, and let the computer go to work."

5. Methods Section, Statistical Methods, Second Paragraph: The authors use ROC-AUC to assess their model discrimination, which is good. However, their calibration measures (sensitivity, specificity, PPV, NPV) are not adequate for assessing model calibration. Van Calster (2016) actually refers to these measures as "mean calibration", which is a step below "weak calibration". The authors are strongly encouraged to add a calibration plot of observed vs. predicted risk, which more definitively shows the quality of model calibration. Otherwise, the authors need to explicitly state, in their abstract, methods, results and discussion, that their model is not calibrated, even in the weak sense.

6. Results, Table 2: It would be more clear if the authors referred to their models as Vitals-Only and Vitals+labs. Reporting Vitals vs. Labs makes it sound like one model contains only lab measures and the other only vital measures, which is not the case.

7. Results: Only model discrimination is discussed, as only AUC is elaborated upon. No model calibration measures (sensitivity, specificity, PPV, NPV) are mentioned in the results. This is troubling, as the PPV for this model is abysmally low; less than 5% in most cases. As such this model would lead to more false positives than true positives at a rate of at least 19:1. No false positives might be better than false negatives with respect to cardiac arrest, but the authors make no mention of this in the results.

8. Discussion, Limitations, Sixth Paragraph: It is hear that the authors state that they focused on discrimination rather than calibration. They confusingly make a general comment stating that calibration was "unfavorable to our research design," since calibration is "highly unstable as opposed to model discrimination." This is simply unacceptable. The biomedical informatics field has been consistently clear that uncalibrated models are worthless, and both discrimination and calibration for diagnostic and prognostic models are required, especially in supervised learning scenarios such as this. Both the Journal of Biomedical Informatics and the Journal of the American Medical Informatics Association are filled with articles (see anything by Royston, Moons, or Van Calster) stating the importance of calibrated models; they also have numerous articles showing HOW to calibrate.

6. PLOS authors have the option to publish the peer review history of their article (what does this mean?). If published, this will include your full peer review and any attached files.

Reviewer #1: No

Reviewer #2: No

---

## [Author Response · Author response to Decision Letter 0]

19 May 2020

14 May 2020

Dr Joerg Heber

Editor-in-Chief

PLOS ONE

Dear Dr Heber:

Thank you for your ongoing consideration of our manuscript for publication in PLOS ONE. We appreciate the time spent by you and the reviewers and believe the revised manuscript has been improved. Below, we have addressed the reviewers’ comments.

We look forward to your editorial decision.

Sincerely,

Ryo Ueno

On behalf of the authors.

Australian and New Zealand Intensive Care Research Centre, School of Public Health and Preventive Medicine, Monash University, Melbourne, Australia

Email: ryo.ueno@monash.edu

Reviewer #1: 

General Comment: 

The topic of this article is important however substantial revisions are suggested to improve the quality and clarify of the work. Overall the language has to be improved. Additionally, authors have often used uncommon words such as “parsimonious” , “ plausible” , etc — it is suggested to use simpler words and sentences. Although the meaning remains the same , a simple sentence/word is less likely to be misinterpreted and is also well received by international readers.

[Response]

We appreciate your comments. As suggested, we have modified the words and sentences in all sections. In addition, we have sought help from a professional language editing service for this purpose. 

Specific Comments:

1. Introduction: it is too short and needs major changes. The research gap is not well supported. It is suggested to clarify the state of the art, the motivation, and the research question.

[Response]

As requested, we have changed the Introduction section (page 4, para 1-3) to address the following topics in separate paragraphs: the state of the art, research gap, motivation, and the research question. 

Overall, our biggest motivation for this study was to compare the additional value of laboratory results using the same dataset and same algorithm. Prior studies use various datasets with various algorithms for the prediction of IHCA, and a comparison of such studies is unable to clarify whether laboratory results are really necessary in the prediction of IHCA. 

2. Method: the subsection “outcome” might confuse readers. Please assign a different subsection name. Also provide with a reasoning for the choice of your ML model. It is also suggested to list the limitations of the selected model and address them in the paper.

[Response]

We thank the reviewer for this helpful suggestion. First, as requested, we have changed the subsection name from “outcome” to “prediction outcome measures.” in the Methods section (page 7, para2). Second, as suggested, we have summarized three reasons why we chose the random forest model. The limitations of this algorithm are briefly summarized in the Methods section (page 7, para3), and further discussed in the Discussion section (page 16, para 3). 

3. Discussion: it required major rework. Not only English but also the flow has to be improved. Many claims need citations. I have highlighted some of them :

“Prior studies have extensively investigated the importance of early prediction of IHCA.” [cite]

“Clinical deterioration is common prior to the cardiac arrest [4,5] and monitored or witnessed patients have more favorable outcomes [2,3] “ — this sentence is very confusing. I suggest rephrasing it.

“ Historically, multiple early warning scores have been developed, ranging from an analogue model based on vital signs to a digital scoring system using both vital signs and laboratory resu...” [cite]

“ Similar results were obtained in other studies [28]. ” — cite more studies or rephrase the sentence (studies — study) .

[Response]

As suggested, we have added a number of citations. In addition, we have restructured the discussion with input from all the authors and sought the help of a professional language editing service to refine the text. 

4. In the discussion I also suggest to include subsections discussing the effects biases in such ML models and how this has been addressed in the study. What are the risks associated?. Is the outcome clinically meaningful? (Please read the following to build upon discussion section: https://doi.org/10.12968/bjhc.2019.0066 ; https://doi.org/10.1126%2Fscience.aaw0029)

[Response]

We thank the reviewer for these insightful comments on the bias inherit to ML models. We have read the citations with great interest and respond to each issue below. 

1) Bias and risks inherent to this model

First, this model was developed in a single-center setting. External validation with other clinical setting with other demographic will be of great importance. For clinicians, we have prepared Table 1 to detail the background of our cohort. 

Second, we appreciate that we did not provide any clinical practice as a benchmark comparison for the Vitals-only data. However, this study focuses on the comparison of the Vitals-Only model and Vitals+Labs model rather than on providing a single model for clinical use. It would be essential to provide appropriate benchmarks before assessing a model’s clinical utility, but that is outside the scope of this paper. 

Third, we have yet to assess whether our models are interoperable. Theoretically, our model only uses commonly collected variables and is both generalizable and interoperable. Future investigation should include actually implementing the random forest based model in an electronic medical record. 

These limitations are now summarized in the Limitations section (page 15, para1; page 17, para2-3). 

2) Validity of the outcome

As stated above, we have read the suggested articles with great interest. We noticed that there are a few definitions of “outcome” in those articles. Here, we discuss three outcomes related to our research.

i) Outcome, as in the performance measure 

In our study, we used not only the AUC but other commonly used statistical values such as sensitivity, specificity, positive (negative) prediction value, and positive (negative) log-likelihood. In addition, we have created a calibration plot in the modified manuscript. In this plot, the x-axis summarizes the predicted probability whereas the y-axis shows the observed proportions of patients who have a cardiac arrest in next 8 hours. In this plot, almost 20%–30% of the patients had an IHCA if their predicted risk score was higher than 0.9. These results are summarized in the Results section (page 10, para 2; page 11, para 1). 

ii) Outcome, as in the endpoint of the prediction

The aim of our models is to predict unexpected IHCA rather than a combined outcome or surrogate outcome. Unlike other outcomes such as ICU transfer, unexpected IHCA is always objective and always necessitates clinical intervention. This is now described in the Discussion section (page 14, para 2). 

iii) Outcome, as in the patient-centered outcome as a result of implementing this model 

We have yet to determine whether our prediction model might actually improve the trajectory of patients at risk. However, our results are persuasive enough to facilitate a prospective validation of the Vitals-Only model. We describe this in the Discussion section (page 17, para 3).

Reviewer #2: 

General Comment: 

The authors present their findings from a diagnostic model development project, aiming to compare two machine learning algorithms for predicting in-hospital cardiac arrest: a standard approach using both vital sign and laboratory measures from an EHR, and a reduced model using only the vital sign components. They report that both models provide similar discrimination in-hospital cardiac arrest occurrence, with similar results in several settings. The debilitating omission in this manuscript is that authors provide only the crudest of calibration/validation measurements, and double down on their refusal to calibrate in their Discussion. The use of random forest does not lead to a generalizable model, at least how used here.

[Response]

We thank the reviewer for these insightful suggestions. As suggested, we have added a calibration analysis. In addition, we have described both the reasons for choosing and limitations of using a random forest algorithm in our model. Below, we elaborate on both topics in detail.

Specific Comments:

1. The Introduction and Methods section combined are shorter than the Discussion. As a result there is little motivation for the project and the methods used by the authors are not reported in enough detail, particularly the analytic approach (more on both of these below).

[Response]

As suggested, we have clarified the motivation for the project in the Introduction section (page 4, para 3). In addition, we have restructured the Introduction with paragraphs on the following topics: the state of the art, the research gap, motivation, and the research question. (page 4, para 1-3; page 5, para 1)

Overall, our biggest motivation for this study was to compare the additional value of laboratory results using the same dataset and same algorithm. Prior studies use various datasets with various algorithms for the prediction of IHCA, and comparison of such studies is unable to clarify whether laboratory results are necessary in the prediction of IHCA.

We also address the concerns you mention regarding our methods in the comments below. 

2. Introduction, Second paragraph: The authors main justification of why they would like to investigate omitting laboratory measurements is that the effect of doing so is unknown. While true, this is hardly justification for why they should feel doing so will not affect the diagnostic tool. There has to be a conceptual reason as to why they think this would work, and that reason(s) needs to be reported.

[Response]

As the reviewer points out, our initial draft lacked a conceptual reason as to why the Vitals-only model may have a similar performance to that of Vitals+Labs model. As summarized in the modified Introduction (page 4, para 2), we believe that vital signs are more immediately available for all the patients regardless of the clinical settings compared to laboratory results. Also, vital signs may reflect the acute physiological change in human body better than change in electrolytes or proteins in the blood. In addition, the amount of data will be much larger for the vital signs, as they can be easily collected any time. Unlike laboratory results, which are usually taken twice or thrice weekly, the amount of data of vitals signs will be larger. Thus, we believe it important to investigate if Vitals-Only model could perform as well as Vitals+Labs model. 

3. Methods Section, Study Population: The authors report that the vital signs and laboratory results are shown in detail in Figure 1. However, this figure shows nothing of the sort.

[Response] 

We mistakenly added this information to the caption of the Figure 1. We have now changed Figure 1 to provide this content. 

4. Methods Section, Statistical Methods: The authors use the random forest model for the prognostic tool since -- as they state -- it usually outperforms other algorithms. The reason for that is because it leads to models of great complexity, and model-averages across numerous well-fitting models. This leads to an uninterpretable model, which limits the ability for anyone (let alone diagnostic clinicians or scientists) who might want to use it. Because of this method, the authors do not -- and could not even if they wanted to -- report any prognostic model. As such, there approach is not reproducible and appears of little value then "use random forests, add these vital sign measurements, and let the computer go to work."

[Response]

We appreciate the reviewer’s comments and the opportunity to clarify this important point.

As the reviewer points out, the random forest model is uninterpretable. As a result, we are unable to determine why our model predicts an IHCA for a particular patient. Hence, the interpretation of the model’s results may differ among medical personnel and thus are not reproducible. 

Such a drawback, however, may not be an insurmountable obstacle in the clinical application of this model for the following three reasons. 

First, a clinician’s response to this alert is likely to be quite straightforward regardless of the reason for the alert. Clinicians will review the patient, collect both electronic and bedside information to assess the situation, and will escalate the care as appropriate. There are variety of interventions that could be made such as additional testing, fluid administration, or transfer to the ICU. However, what determines the intervention is not an alert per se but the clinician’s review of the situation. As long as the early warning system triggers such action of the clinician, the reason behind the alert might be of marginal importance. The clinician can then work out an appropriate response based on the patient’s features rather than the model’s feature.

Second, various de facto standards in our usual medical practice are full of “black boxes.” Not all clinicians are aware of the mechanisms behind MRI. Not all clinicians understand why a certain saddle shape of an electrocardiogram is highly associated with a fetal congenital arrhythmia, but all emergency physicians utilize this knowledge in their daily practice. “Prepare the ECG machine, add electrodes, and let the machine prepare the waveform” is often what actually happens at the bedside. Likewise, the fact that this model is uninterpretable does not always mean it is not useful at the bedside. 

Third, the random forest model is in fact partially interpretable. Of course, we are unable to know why a patient was predicted to have IHCA (or otherwise), but we can at least know the weight of each variable in the feature matrix. This at least enables us whether the decision-making algorithm is consistent with our clinical intuition.

We appreciate the insightful comments of the reviewer and added the above discussion to the Discussion section. (page 16, para 4; page 17, para 1)

5. Methods Section, Statistical Methods, Second Paragraph: The authors use ROC-AUC to assess their model discrimination, which is good. However, their calibration measures (sensitivity, specificity, PPV, NPV) are not adequate for assessing model calibration. Van Calster (2016) actually refers to these measures as "mean calibration", which is a step below "weak calibration". The authors are strongly encouraged to add a calibration plot of observed vs. predicted risk, which more definitively shows the quality of model calibration. Otherwise, the authors need to explicitly state, in their abstract, methods, results and discussion, that their model is not calibrated, even in the weak sense.

[Response]

We strongly agree with the reviewer. As suggested, we have added a calibration plot in Figure 3. We discuss the interpretation of the result in the Results (page 11, para 1) and Discussion (page 16, para 3) sections. 

6. Results, Table 2: It would be more clear if the authors referred to their models as Vitals-Only and Vitals+labs. Reporting Vitals vs. Labs makes it sound like one model contains only lab measures and the other only vital measures, which is not the case.

[Response]

As suggested, we have changed the names of each model to Vitals-Only and Vitals+Labs throughout the manuscript. 

7. Results: Only model discrimination is discussed, as only AUC is elaborated upon. No model calibration measures (sensitivity, specificity, PPV, NPV) are mentioned in the results. This is troubling, as the PPV for this model is abysmally low; less than 5% in most cases. As such this model would lead to more false positives than true positives at a rate of at least 19:1. No false positives might be better than false negatives with respect to cardiac arrest, but the authors make no mention of this in the results.

[Response]

As suggested, we added a calibration plot in Figure 3. We will discuss this plot in the next response. As the reviewer points out, the model has a low positive predictive value, and this is a global issue in all prediction models of rare events such as IHCA (0.4% of prevalence)[1][2]. Although no false positives might be better than false negatives with respect to cardiac arrest, each hospital needs to seek its own balance to optimize the tradeoff between false alarms and overlooked IHCA patients when implementing our model in clinical practice. 

In this study, we focused on the comparison of the Vitals-Only model and Vitals+Labs model at a widely investigated threshold (i.e., Youden’s index) rather than the provision of a highly predictive model. Therefore, we did not aim for overly sensitive model nor an overly specific model. A different threshold and more sophisticated feature engineering might aid in developing a model with a higher positive predictive value. 

Of note, it is also worth investigating the outcomes of “false positive” patients in future studies. Patients flagged as being at risk of cardiac arrest, even if the prediction is a “false positives” may be still identified as “at risk” patients, which would facilitate early interventions such as those of a rapid response team. 

We have added this discussion in the Results (page 10, para 2) and Discussion section (page 15, para 4; page 16, para 1). 

8. Discussion, Limitations, Sixth Paragraph: It is here that the authors state that they focused on discrimination rather than calibration. They confusingly make a general comment stating that calibration was "unfavorable to our research design," since calibration is "highly unstable as opposed to model discrimination." This is simply unacceptable. The biomedical informatics field has been consistently clear that uncalibrated models are worthless, and both discrimination and calibration for diagnostic and prognostic models are required, especially in supervised learning scenarios such as this. Both the Journal of Biomedical Informatics and the Journal of the American Medical Informatics Association are filled with articles (see anything by Royston, Moons, or Van Calster) stating the importance of calibrated models; they also have numerous articles showing HOW to calibrate.

[Response]

We thank the reviewer for these insightful comments. As suggested, we have created a calibration plot. As shown in Figure 3, the calibration performance of the two models does not differ much. The highest risk group has a 20%–30% occurrence of IHCA, which is far more frequent than its pre-test probability among the entire population. However, the calibration plot is far from the diagonal line with poor calibration score (Vitals-Only model vs Vitals+Labs model, Hosmen-Lemeshow C-statistics = 8592.40 vs 9514.76, respectively). Therefore, we should not interpret the predicted probability from our models as an observed proportion of patients having IHCA in the next 8 hours.

The reason for this calibration performance may stem from the design of our model. The aim of this research is a comparison of the discrimination performance of two models rather than a provision of a well-calibrated risk score. Therefore, we attempted to maximize its discrimination score at the expense of calibration performance. However, if we were to implement the Vitals-Only model in an actual clinical setting, a well-calibrated model might be highly influential on clinical decision making by enabling clinicians to interpret the predicted probability as a risk score. Although some clinical tests are still solely focused on discrimination (e.g., pregnancy tests, fecal occult blood test), they should ideally all be calibrated. Future studies should investigate a more sophisticated calibration of the Vitals-Only model before its clinical application. 

Regardless, we believe that the result of our study shows that the simpler Vitals-Only model performs as good as Vitals+Labs model, and this result will be of value in the further development of IHCA prediction models

We have added the above discussion in the Discussion section (page 16, para 3). 

Figure 3: Calibration plot

The x-axis summarizes the predicted probability of having a cardiac arrest in the next 8 hours, whereas the y-axis shows the observed proportions of patients who have a cardiac arrest in next 8 hours. Histogram below the calibration plot summarizes the distribution of predicted probability amongst all the patients in our dataset. 

Abbreviations: IHCA; in-hospital cardiac arrest 

REFERENCES

[1] Goto T, Camargo CA, Faridi MK, Freishtat RJ, Hasegawa K. Machine Learning–Based Prediction of Clinical Outcomes for Children During Emergency Department Triage. JAMA Netw Open 2019;2:e186937. doi:10.1001/jamanetworkopen.2018.6937.

[2] Haibo He, Garcia EA. Learning from Imbalanced Data. IEEE Trans Knowl Data Eng 2009;21:1263–84. doi:10.1109/TKDE.2008.239.

---

## [Decision Letter · Decision Letter 1]

2 Jun 2020

PONE-D-20-03952R1

Value of Laboratory Results in Addition to Vital Signs in a Machine Learning Algorithm to Predict In-Hospital Cardiac Arrest: A Single-Center Retrospective Cohort Study

PLOS ONE

Dear Dr. Ueno Ryo,

Thank you for submitting your manuscript to PLOS ONE. After careful consideration, we feel that it has merit but does not fully meet PLOS ONE’s publication criteria as it currently stands. Therefore, we invite you to submit a revised version of the manuscript that addresses the points raised during the review process.

ACADEMIC EDITOR: Our expert reviewer(s) have recommended additional revisions to your revised manuscript. There are many area that need clarifications and improvement. Therefore, I invite you to respond to the reviewer(s)' comments as listed and revise your manuscript.

We look forward to receiving your revised manuscript.

Kind regards,

Wisit Cheungpasitporn, MD, FACP

Academic Editor

PLOS ONE

Additional Editor Comments:

Our expert reviewer(s) have recommended additional revisions to your revised manuscript. There are many area that need clarifications and improvement. Therefore, I invite you to respond to the reviewer(s)' comments as listed and revise your manuscript.

Reviewers' comments:

Reviewer's Responses to Questions

**Comments to the Author**

1. If the authors have adequately addressed your comments raised in a previous round of review and you feel that this manuscript is now acceptable for publication, you may indicate that here to bypass the “Comments to the Author” section, enter your conflict of interest statement in the “Confidential to Editor” section, and submit your "Accept" recommendation.

Reviewer #1: (No Response)

Reviewer #2: All comments have been addressed

2. Is the manuscript technically sound, and do the data support the conclusions?

Reviewer #1: Yes

Reviewer #2: (No Response)

3. Has the statistical analysis been performed appropriately and rigorously? 

Reviewer #1: Yes

Reviewer #2: (No Response)

4. Have the authors made all data underlying the findings in their manuscript fully available?

Reviewer #1: Yes

Reviewer #2: (No Response)

5. Is the manuscript presented in an intelligible fashion and written in standard English?

Reviewer #1: No

Reviewer #2: (No Response)

6. Review Comments to the Author

Reviewer #1: The authors have made substantial revisions. However, more work is needed:

1. I understand the motivation of the work, however, just stating that vitals are readily available. I suggest elaborating on the "data dimensionality," "minimal optimal" problems, etc. emphasizing the benefits of a smaller or optimal dataset. Additionally, did the computational time reduced significantly after eliminating predictors? It would be better if authors can report the computation time (for their computer configuration).

2. "A simpler model with fewer input ... " I do not agree. Especially in a healthcare setting, the adoption of any technology takes place based on its performance and economic feasibility.

3. "... it does not require any log-transformation .... computational complexity of the pre-processing is reduced ....". Log transformation and normalization are not computationally complex. I do not suggest stating this as a reason to justify RF.

4. k-fold cross-validation is always preferred to manually divide data into test and train set.

5. "Implausible values were removed ...". This is not clear. What is "implausible values"? Outlier or anomalies? Please rewrite and explain, just giving citations is not sufficient

6. In the STRENGTH section of the paper, the authors stated that " A prior study showed that the existence of missing data itself has predictive value [27]" Here is the quote from the study [27] "However, missing and incorrect demographic information in both the Death Master File and EHR data can affect the accuracy of the matches and the resulting estimated survival rates. To circumvent these limitations, our outcome was literally whether the EHR indicates that the patient is alive three years after our cohort period ended. We were not modeling time until death or conducting a traditional survival analysis." The study [27] did not replace missing values to binary. Also, the standard practice is to delete the column when more than 50% is missing. Missing data can only be indicative of "non-essential information" (if it is deliberately not captured).

7. The study has 3 strengths and nine limitations.

8.The implication of the study is unclear. How eliminating data make an ML model simpler? The model's complexity is independent of data; it depends on the underlying algorithm.

9.Lastly, the language needs to be improved (tense). The sentence structures are not indicative of technical writing. " Second, clinically, doctors and nurses might intervene in patients ... "

Reviewer #2: (No Response)

7. PLOS authors have the option to publish the peer review history of their article (what does this mean?). If published, this will include your full peer review and any attached files.

Reviewer #1: Yes: Avishek Choudhury

Reviewer #2: No

---

## [Author Response · Author response to Decision Letter 1]

19 Jun 2020

Reviewer #1: 

The authors have made substantial revisions. However, more work is needed:

[Response]

We thank the reviewer for the encouraging comment and very constructive suggestions. We provide point-by-point responses below. 

1. I understand the motivation of the work, however, just stating that vitals are readily available. I suggest elaborating on the "data dimensionality," "minimal optimal" problems, etc. emphasizing the benefits of a smaller or optimal dataset. Additionally, did the computational time reduced significantly after eliminating predictors? It would be better if authors can report the computation time (for their computer configuration).

[Response]

Thank you for the constructive suggestions. As suggested, we have further elaborated on the motivation of this study in the Introduction section as follows: 

Fourth, from a computational point of view, the minimal optimal model with a low data dimensionality is always better than a more complicated model as long as it has similar performance.

As the reviewer has pointed out, the computational time is an important aspect of a machine learning model. In our study, however, the reduction in dimensions was only 117 to 49, and we did not observe any significant reduction in computation time. 

Despite the lack of computational benefit, we believe achieving a minimal optimal model is of clinical importance. The bedside application of a ML-model always comes with the cost of data acquisition; every single laboratory test is a physical and financial burden for both hospital staff and patients. If the performance of a Vitals-Only model is not inferior to that of the Vitals+Labs model, it is more economically feasible. 

2. "A simpler model with fewer input ... " I do not agree. Especially in a healthcare setting, the adoption of any technology takes place based on its performance and economic feasibility.

[Response]

We agree with the reviewer. We believe that Vitals-Only model will be more economically feasible than the Vitals+Labs model because laboratory tests have physical and financial costs. As our results show, the performance is comparable despite its simplicity. This was our reasoning, which we agree we did not make explicit in this sentence. We have edited the Introduction section as follows: 

Third, Vitals-Only model would be non-invasive and economically feasible, because it does not require any laboratory tests, which are both physically and financially stressful for patients.

We also changed the sentence you mention in your comment as follows.

First, a simpler model with fewer input variables can be adopted in a wide variety of settings.

  

3. "... it does not require any log-transformation .... computational complexity of the pre-processing is reduced ....". Log transformation and normalization are not computationally complex. I do not suggest stating this as a reason to justify RF.

[Response]

We thank the reviewer for this comment. We have deleted the sentence as suggested. 

4. k-fold cross-validation is always preferred to manually divide data into test and train set.

[Response]

We appreciate the reviewer’s insightful suggestion. We chronologically defined the training period and test period a priori. We used this approach for the following reasons. First, most of our variables are time dependent; vital signs and laboratory results change over time. In addition, the information about the previous episode of in-hospital cardiac arrest is also time dependent. Second, when we actually implement the model in an actual clinical setting, we need to develop a model using data from a certain period of time and prospectively apply it to data from a distinct and more recent period. As such, our approach may be a proxy for such a situation. 

5. "Implausible values were removed ...". This is not clear. What is "implausible values"? Outlier or anomalies? Please rewrite and explain, just giving citations is not sufficient

[Response]

We mean data that is not biologically possible, which indicates that an error in data entry has occurred. We have changed this term in the text to ‘values that are clearly errors’. We agree with the reviewer that more clarification is needed. All the biologically inconsistent data that were excluded are summarised in the Supplementary Information as follows: 

List of excluded inconsistent data 

1. body temperature < 30 ℃ or > 45 ℃

2. heart rate <20 beats/min or >300 beats/min

3. respiratory rate >80/min

4. systolic blood pressure < 20 mm Hg or > 300 mm Hg

5. diastolic blood pressure < 10 mm Hg or > 200 mm Hg

6. urine output > 10,000 ml per eight hours

7. oxygen saturation < 40% or > 101%

 

6. In the STRENGTH section of the paper, the authors stated that " A prior study showed that the existence of missing data itself has predictive value [27]" Here is the quote from the study [27] 

"However, missing and incorrect demographic information in both the Death Master File and EHR data can affect the accuracy of the matches and the resulting estimated survival rates. To circumvent these limitations, our outcome was literally whether the EHR indicates that the patient is alive three years after our cohort period ended. We were not modeling time until death or conducting a traditional survival analysis." 

The study [27] did not replace missing values to binary. Also, the standard practice is to delete the column when more than 50% is missing. Missing data can only be indicative of "non-essential information" (if it is deliberately not captured).

[Response]

Thank you for the meticulous comment and the opportunity for further clarification. 

We agree with the reviewer; the literature [27] assessed the predictive value of the time when each laboratory test was ordered, compared with the predictive value of the result of each laboratory result. As such, this study is not about missing data imputation. We do believe, however, that this study proved the importance of the time and frequency of each test (including the decision NOT to measure) in predicting adverse outcomes. Hence, we aim to incorporate the timing and frequency of each measurement by changing each variable to a binary value. We note that the limitation quoted by the reviewer was about the demographic information, not clinical tests. 

To ensure this issue is addressed, we have repeated the primary analysis with four different methods of handling missing values. All the results were similar to the primary analysis, as summarized in Supplementary Table 3. We believe that this sensitivity analysis reinforces the robustness of the primary outcome. To avoid confusion, we have changed the text in the Discussion section. 

Supplementary Table 3: Predictive Performance of Each Model for the In-hospital Cardiac Arrest with Various Imputation Methods

 Imputation methods

IHCA Imputation Binary Deletion Categorical

0-8h Vitals-Only 0.877 0.896 0.887 0.899

0-8h Vitals+Labs 0.855 0.864 0.848 0.862

0-24h Vitals-Only 0.851 0.871 0.850 0.873

0-24h Viltas+Labs 0.828 0.849 0.828 0.846

Missing values were imputed with the most recent variable value or average of the overall sample if <50% of the values were missing. Otherwise, a variable was converted according to the following rules:

1) Imputation (the variable was imputed with the most recent variable value or average of the overall sample);

2) Binary (if the variable was missing it was converted to 0, otherwise it was set to 1);

3) Categorical (the variable was converted to a categorical value, i.e. missing or quantile category);

4) Deletion (the variable was removed from the entire analysis).

7. The study has 3 strengths and nine limitations.

[Response]

We have noticed that Limitations section includes both limitations and future research directions. We have changed the subheading to better reflect the content of the paper. 

8.The implication of the study is unclear. How eliminating data make an ML model simpler? The model's complexity is independent of data; it depends on the underlying algorithm.

[Response]

We thank the reviewer for this insightful comment. 

We agree that improving the underlying algorithm is essential to creating a faster and more accurate model. We also agree that the complexity of the dataset is unimportant in a retrospective setting, when a dataset is readily available. When we apply such a model at the bedside, however, we always need to collect data prospectively, and such data is full of missing variables for the following reasons: 

1. We cannot justify collecting laboratory results every day, just for the purposes of prediction. Collecting laboratory results at every opportunity is both a physical and financial burden on patients. 

2. Even if we take a blood sample every day, we rarely order all types of blood test for every single patient. For some patients, just checking the hemoglobin is enough for his/her management, and we cannot justify ordering all blood tests. 

3. Even if we were able to order all the blood test results, a couple of hours are needed before those results become available to the clinicians. 

Therefore, a model with vital signs that is readily available and does not place any additional burden on patients is highly desirable in practice. 

9.Lastly, the language needs to be improved (tense). The sentence structures are not indicative of technical writing. " Second, clinically, doctors and nurses might intervene in patients ... "

[Response]

We appreciate the reviewer’s suggestion. As suggested, we have edited the language with a specific focus on tense and structure. Furthermore, we used an additional language editing service.

---

## [Decision Letter · Decision Letter 2]

24 Jun 2020

Value of Laboratory Results in Addition to Vital Signs in a Machine Learning Algorithm to Predict In-Hospital Cardiac Arrest: A Single-Center Retrospective Cohort Study

PONE-D-20-03952R2

Dear Dr. Ryo,

We’re pleased to inform you that your manuscript has been judged scientifically suitable for publication and will be formally accepted for publication once it meets all outstanding technical requirements.

Kind regards,

Wisit Cheungpasitporn, MD

Academic Editor

PLOS ONE

Additional Editor Comments:

I reviewed the revised manuscript and the response to reviewers' comments. Revised Manuscript is well written. All comments have been addressed and thus accepted for publication.

Reviewers' comments:

Reviewer's Responses to Questions

**Comments to the Author**

1. If the authors have adequately addressed your comments raised in a previous round of review and you feel that this manuscript is now acceptable for publication, you may indicate that here to bypass the “Comments to the Author” section, enter your conflict of interest statement in the “Confidential to Editor” section, and submit your "Accept" recommendation.

Reviewer #1: All comments have been addressed

Reviewer #3: All comments have been addressed

2. Is the manuscript technically sound, and do the data support the conclusions?

Reviewer #1: Yes

Reviewer #3: Yes

3. Has the statistical analysis been performed appropriately and rigorously? 

Reviewer #1: N/A

Reviewer #3: Yes

4. Have the authors made all data underlying the findings in their manuscript fully available?

Reviewer #1: Yes

Reviewer #3: Yes

5. Is the manuscript presented in an intelligible fashion and written in standard English?

Reviewer #1: Yes

Reviewer #3: Yes

6. Review Comments to the Author

Reviewer #1: (No Response)

Reviewer #3: All concerns have been fully elucidated, missing sections and analyses have been completed. Finally, comprehension errors have been corrected. Good work!

7. PLOS authors have the option to publish the peer review history of their article (what does this mean?). If published, this will include your full peer review and any attached files.

Reviewer #1: Yes: Avishek Choudhury — Stevens Institute of Technology

Reviewer #3: No

---

## [Editor Report · Acceptance letter]

29 Jun 2020

PONE-D-20-03952R2 

Value of Laboratory Results in Addition to Vital Signs in a Machine Learning Algorithm to Predict In-Hospital Cardiac Arrest: A Single-Center Retrospective Cohort Study 

Dear Dr. Ueno:

I'm pleased to inform you that your manuscript has been deemed suitable for publication in PLOS ONE. Congratulations! Your manuscript is now with our production department. 

Kind regards, 

on behalf of

Dr. Wisit Cheungpasitporn 

Academic Editor

PLOS ONE